# The Interaction between Flavonoids and Intestinal Microbes: A Review

**DOI:** 10.3390/foods12020320

**Published:** 2023-01-09

**Authors:** Hui-Hui Xiong, Su-Yun Lin, Ling-Li Chen, Ke-Hui Ouyang, Wen-Jun Wang

**Affiliations:** 1College of Food Science and Engineering, Jiangxi Agricultural University, Nanchang 330045, China; 2College of Animal Science and Technology, Jiangxi Agricultural University, Nanchang 330045, China

**Keywords:** flavonoids, gut microbiota, metabolic mechanism, interaction, diseases

## Abstract

In recent years, research on the interaction between flavonoids and intestinal microbes have prompted a rash of food science, nutriology and biomedicine, complying with future research trends. The gut microbiota plays an essential role in the maintenance of intestinal homeostasis and human health, but once the intestinal flora dysregulation occurs, it may contribute to various diseases. Flavonoids have shown a variety of physiological activities, and are metabolized or biotransformed by gut microbiota, thereby producing new metabolites that promote human health by modulating the composition and structure of intestinal flora. Herein, this review demonstrates the key notion of flavonoids as well as intestinal microbiota and dysbiosis, aiming to provide a comprehensive understanding about how flavonoids regulate the diseases by gut microbiota. Emphasis is placed on the microbiota-flavonoid bidirectional interaction that affects the metabolic fate of flavonoids and their metabolites, thereby influencing their metabolic mechanism, biotransformation, bioavailability and bioactivity. Potentially by focusing on the abundance and diversity of gut microbiota as well as their metabolites such as bile acids, we discuss the influence mechanism of flavonoids on intestinal microbiota by protecting the intestinal barrier function and immune system. Additionally, the microbiota-flavonoid bidirectional interaction plays a crucial role in regulating various diseases. We explain the underlying regulation mechanism of several typical diseases including gastrointestinal diseases, obesity, diabetes and cancer, aiming to provide a theoretical basis and guideline for the promotion of gastrointestinal health as well as the treatment of diseases.

## 1. Introduction

The human gut tract harbors a diverse microbial community [1], composed of beneficial flora, harmful flora and neutral bacteria. It is apparent that intestinal microbiota convolves with human beings [2]. It is involved in human metabolism and affects the health of the host. The interactions among intestinal microbiota and metabolism have an impact on human health and disease. Indeed, the disruption of the intestinal microbiota contributes to intestinal diseases such as inflammatory bowel disease (IBD), systematic diseases such as diabetes, etc. [3] The gut microbiota is modified by various factors, including intrinsic factors such as changes in the functional structure and extrinsic factors such as medications and diets in flavonoids that have been implicated as treating the diseases [3].

Nowadays, in response to medical discoveries, an increasing number of studies on nonnutritive bioactive components in plants have been demonstrated to regulate the gut microbiota and further prevent and treat diseases. Thus, it is necessary to place more emphasis on microbiota-flavonoid bidirectional interactions for better understanding the effects of flavonoids. In an attempt to elucidate the mechanisms involved, the role of different flavonoids on host health through gut microbes is frequently studied in animal models, whereas little is known about the specific metabolic pathways of flavonoids by gut microbiota. Moreover, the regulation of diseases through the microbiota-flavonoid interaction remains unclear due to the heterogeneity of the flavonoids and the complexity of the human gut microbiota [4]. In this review, we summarize the current status of the effects of flavonoids on intestinal microorganisms and the mechanisms of flavonoids on disease regulation through intestinal flora with an emphasis on the interaction. More importantly, it helps to fill the current research gaps and indicate the future research directions in this area.

## 2. The Classes of Flavonoids

Flavonoids are a large class of compounds based on the C6-C3-C6 skeleton that consists of two benzene rings (A and B) with phenolic hydroxyl, and they are divided into several classes, including isomers of flavonoids and their hydrogenated reduction products, e.g., flavones, flavonols, flavanones, isoflavonoids and anthocyanins [5]. Moreover, many flavonoids shape in plants via hydroxyl, methoxyl, methyl, or glycosyl substitution patterns, most of which present as glycosides or carbon-based groups combining with sugar in vacuoles of flowers, leaves, stems or roots, and parts of which exist in the free form [6]. Pharmacological evidence has revealed that flavonoids display a series of physiological and pharmacological activities not only on the cardiovascular and digestive system, but also on antioxidant, anti-inflammatory, anti-bacterial, anti-viral, anti-cancer, anti-tumor and hepatoprotective functions [7]. There is variability among beneficial effects and bioactivity of flavonoids because of different flavonoid subclasses shown in Table 1 [8,9,10,11,12,13,14,15,16,17,18,19,20,21,22,23,24,25,26,27,28,29,30,31,32]. Flavonoids are metabolized in the human gastrointestinal tract following the oral administration mainly in the liver, and further transformed into new metabolites by microorganisms in the colon. In turn, the composition and abundance of the intestinal microbiota are regulated by the metabolites to fully exert their physiological influence on growing biological activity and bioavailability of flavonoids [12].

Flavonols have a double bond between C2 and C3 and a carbonyl at C4, typically containing hydroxyl or O-glycosides at C3, or glycosides at other positions. The flavonols are occurring widespread in the dicotyledonous plants, as the most common in kaempferol, quercetin and their glycosides such as myricetin. Kawabata et al. [33] revealed that flavonols (galangin, quercetin, and fisetin) enhanced the production of anti-inflammatory substance(s) by *Bifidobacterium* adolescentis and exerted prebiotic actions. According to pharmacological studies, icariin, a prenylated flavonol glycoside isolated from the genus *Epimedium*, has the potential of inhibiting SARS-CoV-2 entry into host cells through modulating the intracellular pathways [34]. It also notably inhibited the activities of food pollutant bacteria. Dias et al. [35] studied *Humiria balsamifera* (Aubl), identified seven flavonoids by high-performance liquid chromatography (HPLC), and showed their antimicrobial potential through in vitro and in vivo assays. Cascaes et al. [36] showed that flavonoids from the methanol extract of Myrcia rufipila mcvaugh leaves (Myrtaceae) had a high inhibitory effect against *Staphylococcus aureus* CCMB 263 and *Staphylococcus aureus* CCMB 285 as well as the yeast *Candida albicans* CCMB 266 using the well diffusion test.

Isoflavones are natural bioactive plant compounds, known as phytoestrogens, which are mainly present in Angiospermae, Leguminosae, Iridaceae and Rosaceae plants with soybeans being the richest source in the free form of glycosides or in conjugation with glycosides [12]. Isoflavone glycosides are hydrolyzed to the aglycone form upon glucosidases and lactase-phlorizin hydrolase prior to absorption in the small intestine [13]. There is scientific evidence showing the positive health benefit of isoflavones on reducing the incidence of many diseases, such as cardiovascular disease [37], breast cancer and osteoporosis [38].

Flavanones are widely distributed in higher plants, mainly in the free form in the Asteraceae, Leguminosae and Rutaceae, such as water soap, cocoa, tea, red wine, fruits and vegetables, mainly including hesperidin, naringin, etc. Regarding the synthesis of flavanones, the most common pathway is the aldol condensation of 2-hydroxyacetophenones with benzaldehydes generating different derivatives of chalcones and flavanones. The latter encompasses an array of aglycones and glycosylated forms through an O-glycosidic linkage. The absorption mechanisms mainly depend on proton-driven active transport prior to the metabolism, of which the first step is the deglycosylation of flavanone glycosides within the intestinal epithelium by human and bacterial enzymes such as β-glucosidase; furthermore, aglycones are glucuronated or sulfated into phenolic acids by intestinal microflora. Then, after absorption, glucuronidation, sulfation and methylation occur in the liver [14]. Medical research has shown that flavanones have a wide range of physiological and pharmacological activities such as antioxidant, anti-inflammatory, anti-microbial and anti-cancer properties [39].

Anthocyanins are naturally occurring molecules belonging to the flavonoid class which are widely distributed in the leaves, flowers, fruits and other parts of plants, especially in berries. Anthocyanins constitute a widespread class of water-soluble plants pigments that are responsible for the blue, red and purple colors of many plant tissues, mainly in the form of glycosides [40]. Anthocyanins are characterized by low absorption through hydrolysis into aglycone under the action of specific enzymes or a special active transport mechanism to transport glycosides across the intestine wall [40]. After their synthesis, anthocyanins are decomposed by colonic microorganisms [41] and further metabolized by the enterohepatic cycle mainly through the following two possible metabolic processes. Phase I metabolism is that the glycoside enters the cells through sodium-dependent glucose transporter 1 (SGLT1) after hydrolysis in the liver via P450 monooxygenases. Phase II metabolism by enzymes such as urinary-5′-diphosphate glucuronosyltransferase (UGT) or catechol-O-methyltransferase (COMT) [42] involves conversion in smaller phenolic compounds or conjugates of the aglycone, in which bacteria play a significant role in the cleavage of glycosidic linkages and breakdown of the heterocycle [41]. Apart from the well-known antioxidant activity, anthocyanins have a wide variety of health-promoting biological activities. In vitro studies and animal experiments have demonstrated that anthocyanins have positive effects on reducing inflammation, lowering lipids [42], anti-cancer [24], anti-diabetic, anti-cardiovascular potential [25], as well as low-grade chronic inflammation prevention [43].

Procyanidins are different from anthocyanins, as procyanidin is a kind of polyphenolic compound that can produce anthocyanin under thermal acid treatment. It is internationally recognized that it is a natural antioxidant that could efficiently scavenge free radicals. It is widely found in the skin, shell and seeds of plants, among which grape seeds have the highest content of anthocyanins. In addition, it has biological activities on promoting blood circulation [44], protecting liver injury [45], etc. Yang et al. found that procyanidins exhibited neuroprotective activities against cerebral ischemia reperfusion injury by inhibiting the TLR4-NLRP3 inflammasome signal pathway [46]. Procyanidins are known and essential modulators of the microbiota. Mutually, various models in vivo and in vitro have confirmed that the bioavailability of native procyanidins is influenced by gut microbiota, but the mechanisms of health benefits of procyanidins have not been explored. The gut microflora may act as a mediate tool to evoke the forming of active metabolites of procyanidins, but still need more studies to prove these results [47].

Flavones are mainly distributed in bryophytes, ferns and mule gymnosperms apigenin, in which lignin and their glycosides are the common ones. Flavones can be metabolized and transformed by reactions such as hydrolysis, oxidation, O-/C-glycosylation and reduction, and further absorbed to achieve therapeutic effects. Pharmaceutical studies have demonstrated that it is a promising efficacious candidate in inhibiting rat lens aldose reductase to prevent cataract formation in diabetes [48] and inhibiting adenosine 3′, 5′-cyclic monophosphate phosphodiesterase activity which possesses efficacy on antiphlogistic, antidiarrheal and anthelmintic in traditional Chinese medicine [49]. Moreover, flavones affect the leukocyte migration to exert profound effects on inflammation and cancer [32]. A typical example is that luteolin regulates the activity of Rho GTPases and reveals an effective decrease in leukocytes through an acute and chronic experimental allergic encephalomyelitis rat model [50]; similarly, apigenin affects the Janus kinase 3 (Jak3) activity [51].

Total flavonoid extraction has many kinds of methods including solvent extraction, ultrasound-assisted, microwave-assisted, enzymatic hydrolysis and so on. Different methods may result in different yields or purification. Liquid chromatography-tandem mass spectrometry (LC-MS/MS) and ultra-high-performance liquid chromatography–quadrupole/time of flight mass spectrometry (UHPLC-Q-TOF-MS/MS) is always used to detect the composition of natural extracts.

Toxicological properties of the flavonoids are also a concern. Acute toxicity tests, bone marrow cell micronucleus tests, mice sperm abnormality tests, and Ames tests are conducted in accordance with the “Technical Standards for Testing & Assessment of Health Food”. Generally speaking, flavonoids are safe at the recommended dose, which provides a theoretical basis for developing healthy food or medicine. Guo et al. showed that Robinia pseudoacacia cv. Idaho (RPTF) was safe at the medium dose level from the in vivo toxicity experiments showing that there are no signification differences in the evaluation index of body weight, food intake, organ coefficient and histoanalysis. Therefore, the dose is crucial to the usage of flavonoids [52].

## 3. Intestinal Microbiota

It is estimated that our gut contains up to trillions of microbiota with more than 1000 bacterial species that reside in the human intestine, together with viruses, fungi, protozoa, bacteriophages and other tiny organisms constituting the human body’s microecosystem [1]. Intestinal microbes are basically divided into six dominant phyla: *Firmicutes*, *Bacteroides*, *Proteobacteria*, *Actinomycetes*, *Verrucomicrobia* and *Fusobacteria,* among which *Bacteroides* and *Firmicutes* are the main dominant flora. Under normal circumstances, the community in the intestinal tract is interdependent and mutually restricted in a relatively balanced state, together with intestinal mucosa, becoming a natural barrier to the maintenance of human health. The gut maintains a dynamic equilibrium between beneficial and harmful bacteria to help the host exert various impacts. The microbial core function mainly includes genes encoding glycosaminoglycan degradation, the production of short-chain fatty acids (SCFAs), the biosynthesis of some essential amino acids and vitamins [53]. The functions of intestinal bacteria mainly embody several aspects: influencing host metabolism, intestinal host immunity, and intestinal barrier function. There are a wide variety of host-endogenous and host-exogenous factors that affect the intestinal flora, among which diet emerges as a pivotal determinant in regulating the intestinal microecological balance [54]. Studies have shown that dietary intervention can change microbial gene richness and host metabolism and has an impact on the type and composition of bacteria [55]. Consumption of a particular diet produces different shifts in existing host bacterial genera [56]. The food component which is beneficial to gut health is called prebiotics, which could influence the growth and proliferation of beneficial bacteria in the intestine, contributing to competing for the environment of intestinal proliferation, thereby inhibiting the growth of pathogenic bacteria.

Due to the crucial role of the gut microbiota in human health, when the harmonious symbiosis relationship between humans and bacteria gets disturbed, changes in the number and proportion of bacteria result in intestinal dysbacteriosis, which ultimately leads to various diseases such as obesity, diabetes, cardiovascular disease and non-alcoholic fatty liver disease. Nowadays, the possibility of a link between gut microbiota and diseases deserves great attention. New research methods such as microbial culturomics [1] are being explored although the cause of dysbacteriosis has not been clarified. A deeper understanding of the interaction between the host and gut microbiota is a prerequisite for manipulating the microbiota to promote human health benefits and regulate diseases. There are frontier studies in animal and in vitro models showing that compounds such as phenols play a significant role in the treatment and prevention of diseases [57].

The intestinal microbiota plays a modulatory role in various diseases through metabolism or interaction with the host. The intestinal flora maintains a dynamic balance in the human body. However, balance is susceptible to damage by external factors. Host factors such as diet, ingested drugs, intestinal mucosa, the immune system, and disease status affect the composition of the microbiota. Evidence has shown that impaired intestinal microbiota not only contributes to gut diseases but is inextricably linked to metabolic disorders [58]. The mechanisms underlying intestinal dysbiosis often remain unclear due to multiple variations such as oxidative stress, bacteriophage induction and the secretion of bacterial toxins [59]. There are a wide variety of diseases including intestinal diseases, metabolic disorders and neurological diseases that are associated with intestinal dysbiosis as shown in Table 2 [60,61,62,63,64,65,66,67,68,69,70,71,72,73,74,75,76,77,78,79,80,81,82,83,84,85,86,87].

How to relieve intestinal dysbiosis is primarily taken into consideration. At present, probiotic bacteria are being considered in promising adjuvant treatments for different diseases after the verification of clinical trials and in vivo experiments [88]. Probiotic plays an important role in gut microbiome-associated diseases by shaping the intestinal microbiota. However, more studies should be conducted to elucidate the role of probiotics in the treatment of intestinal diseases.

## 4. Metabolic Effects of Intestinal Microorganisms on Flavonoids

Gut bacteria play an important role in the metabolism of flavonoids, which not only can hydrolyze glycosides, glucuronides and sulfates and perform ring-cleavage, reduction, decarboxylation, demethylation, and dihydroxylation reactions [89], but also helps the bioactivity and bioavailability of flavonoids. The interaction between gut microbes and flavonoids is a result of genomic co-evolution. The enzymes produced by gut microbes and genetically encoded proteins can degrade gut microbes, and microbial fermentation produces end products such as SCFAs, bile acids and choline that are beneficial to host health [90]. Moreover, how microbes metabolize flavonoids can be demonstrated from microbiological, histological and enzymatic perspectives.

### 4.1. The Role of Intestinal Microbiota on the Metabolism of Flavonoids

#### 4.1.1. Metabolism of Flavonoids by Gut Microbiota

Phase I and phase II drug-metabolizing enzymes and efflux transporters in the intestinal tract help to metabolize flavonoids. Flavonoids are selectively absorbed in the gastrointestinal tract after oral administration. Aglycones are highly hydrophobic and capable of being absorbed through the stomach more efficiently, while flavonoid glycosides are not. In contrast to some flavonoids that are absorbed in the small intestine, the unabsorbed flavonoids and their metabolites are hydrolyzed by various hydrolytic enzymes of the intestinal flora into chain catabolites or monomers [91] such as glucuronic acid, sulfuric acid and methylation products. In the small intestine, a small number of glycosides are actively transported through the Na+-dependent glucose transporter (SGLT) of the small intestinal epithelial cell membrane. Then, they are absorbed from the intestinal cavity to the portal vein in the form of glycosides by active transport and carrier-mediated transport. Most flavonoid glycosides can be deglycosylated and hydrolyzed by broad-specific β-glucuronides (BSβG), cytosolic β-glucosidase (CBG) and lactose-phlorizin hydrolase (LPH) to generate flavonoid aglycones, which are passively transported through the epithelial cell membrane. Meanwhile, some biotransformation occurs in enterocytes and hepatocytes. Some polymers can be specifically cleaved into monomers by α-L-rhamnosidase, β-glucosidase and β-glucuronidase, etc., completing the first stage metabolism; these initially metabolized phenolic acids are absorbed. After a while, they reach the liver through the portal circulation, undergoing the second stage of transformation: Aglycones and phenolic acids are combined with glucuronide, sulfate and/or methyl moieties. Afterwards, they are distributed into organs and excreted in urine and forming the hepatic-intestinal cycle [92,93]. Overall, flavonoids and microbial transformations generate a complex metabolic intertwined metabolic network that affects gut microbiota and human health.

Taking the metabolism of the following flavonoids as an example, the fission of the flavonol C-ring is first converted through decarboxylation by breaking the bond between C1 and C2 [94]. Flavanones, occurring as glycosides, first obtain sapogenins by deglycosylation, and methanetriols and 3-propionic acid are obtained by breaking the bond between C1 and C2 or between C3 and C4, whose degraded metabolites are proanthocyanidins [95]. The isoflavone glycosidic hydrolysis of glycine cannot be directly converted to soy sapogenins, which are absorbed on enterocytes through the formation of intermediate dihydro-soy sapogenins to produce estramacol, which undergoes a two-phase reaction glucuronidation [96].

The structure of flavonoids and the types of microorganisms would affect flavonoid degradation by intestinal microorganisms. For instance, the formation of equol depends on the intestinal flora. Germ-free animals are not capable of producing equol, whereas the production of equol requires activated flora such as *Streptococcus intermediums* ssp., *Bacteroides ovatus* spp. Equol and 5-hydroxy-equol are produced by *Lactobacillus fermentum* strains in cow’s milk supplemented with daidzein and genistein [97].

Collectively, flavonoids can be converted into various metabolic catabolites with the effects of hydrolases and dioxygenases present in the microbiota, which play an essential role in human health. The metabolic mechanisms of flavonoids by gut microbiota and impacts on health are shown in Figure 1. Flavonoids have inhibitory effects on relevant diseases by altering the pattern of the intestinal microbiota, and the intake of flavonoids improves the stability of intestinal microbiota by activating SCFAs excretion, intestinal immune function, and other physiological processes [98]. A metabolism study by fecal microflora in vitro suggested that Dihydromyricetin (DMY) could be metabolized by reduction and dehydroxylation metabolic pathways. The influence of DMY on intestinal microbes was studied using 16S rDNA pyrosequencing, which showed that supplementation with DMY was accompanied by the significantly decreasing *Firmicutes* such as *Lactobacillus,* the increasing *Bacteroidetes* as well as the reduced production of SCFAs. It was confirmed in terms of human obesity and glycolipid metabolism that DMY was beneficial to the degradation of polysaccharides and fibers, and digestion and absorption in the gastrointestinal tract [99].

#### 4.1.2. The Role of Gut Microbiota Metabolism and Biotransformation on Flavonoids

Human metabolism is regulated by the interaction between the host and microbial genome. The human gut microbiota encodes a broad diversity of enzymes, thus expanding the metabolic pathways occurring in the human body [100]. Gut microbial xenobiotic metabolites may alter bioactivity, toxicity and bioavailability and interfere with the activities of metabolic enzymes thereby influencing the molecular pathway, which may affect human health. The intake of nutrients and drugs is proved to influence intestinal microbiota metabolism that has noticeable regulatory effects on the metabolic phenotype of the host. The gut microbiota produces bile acids, choline, and SCFAs that are of vital importance to host health [55]. The hydrophobic interactions exist in polyphenols and bile acids. Based on the polyphenol-induced alterations of the bile acid pool, polyphenols are described to significantly change microbiota compositions and affect gene expressions linked to bile acid metabolism [101].

Differences in gut microbiota ecology determine flavonoid metabolism that produces different metabolites, thereby influencing different diseases [102]. Therefore, understanding the inner correlation between gut microbiota metabolism and biotransformation of flavonoids may better illuminate their connection to human health. Gut microbiota metabolites could alter lipoprotein lipase action, very-low-density lipoprotein (VLDL) secretion, and plasma triglyceride levels. It is common that the treatment of obesity and dyslipidemia is attributed to the role of gut microbiota metabolites [64]. Herein, we review our knowledge of how gut microbiota modifies flavonoids and discuss the effects of different metabolic enzymes on flavonoids.

Intestinal microbes are equipped with several enzymes that are specialized in the degradation and biotransformation of flavonoids by enzyme reaction mainly through hydrolysis and reduction reactions. Specific microbial enzymes could also influence gut health by affecting host intestinal metabolites [103]. While the structures of flavonoids are different, the intestinal flora in the human body can degrade flavonoids by encoding different active enzymes through the flavonoid use loci and show different degradation mediators. Several bacteria with β-glucosidase activity are the dominant members of the human intestinal microbiota, including *Bifidobacterium adolescentis, Bifidobacterium longum, Enterococcus faecalis, Bacteroides ovatus, Bacteroides uniformis, Parabacteroides distasonis* and *Escherichia coli*. Flavonols, flavones, flavanones, dihydrochalcones, isoflavones and anthocyanidins are usually deglycosylated by gut bacteria, such as *Bifidobacteriaceae*, *Lactobacillaceae*, *Lachnospiraceae, Enterococcaceae.* In addition, α-L-Rhamnosidases are involved in the deglycosylation of flavonoids, which are characterized from the trains of *Lactobacillus acidophilus*, *Lactobacillus plantarum* and *Bifidobacterium dentium* [104]. Rutin is a glycoconjugate form of quercetin and is poorly absorbed in the small intestine. It is usually hydrolyzed by several gut microbes such as *Lactobacillus acidophilus*, *Lactobacillus plantarum* and *Bifidobacterium dentium* that possess α-rhamnosidase activities. A part of gut microbiota is activated by rutin [105]. Gut microbial enzymes associated with metabolism of flavonoids are detected in the gut microbiota. Enzymes with high sequence similarity can catalyze distinct chemical reactions. Incorporating enzyme discovery and characterization efforts into investigations of gut microbial xenobiotic metabolism can help understand what enzymes are readily and effectively accessible to gut microbiota metabolism. The integration of clinical studies with mechanistic experiments in model systems and organisms helps to understand the gut microbiota metabolizes flavonoids. There are advanced bioinformatic and experimental approaches including protein sequence similarity and genome neighborhood network analyses, high-throughput ligand docking, and structural genomics to associate enzymes with metabolic pathways and substrates [99]. An in-depth understanding of the enzyme encoded by intestinal flora and the mechanism of enzyme structure on the decomposition of flavonoids can promote the development of flavonoids, and provide a theoretical basis for precision medicine targeting intestinal bacteria.

Gut microbiota is responsible for the metabolism of flavonoids into readily absorbed metabolites, which may be responsible for the health effects. The interaction between flavonoids and gut microbiota accelerates the secondary metabolites, such as SCFAs and bile acids. New tools and methods such as metabonomics are widely used to detect related metabolites of gut microbiota. Zhao et al. [106] revealed that the clinical trial of high dietary fiber diet intervention in type 2 diabetes mellitus (T2DM) proved that high dietary fiber degraded H_2_S via intestinal cells and promoted the fat cells uptake of glucose, and increased the enrichment level of SCFAs. In addition, it also had an obvious effect on stimulating the intestinal tract to secrete more Glucagon-like peptide 1 (GLP-1) as well as increasing insulin to reduce inflammation. In summary, high dietary fiber regulated glucose and lipid metabolism to improve diabetes, further indicating that gut microbiota involves in host metabolism. More attention is focused on the influence of intestinal flora on the human body to scratch the regulation and prevention of metabolic diseases in recent years.

### 4.2. Mechanism of Biotransformation and Formation of Flavonoids by Intestinal Microorganisms

Gut microbiota produces a wide range of enzymes that are beneficial to the utilization and biotransformation of flavonoids. In general, some flavonoids cannot be metabolized in the human gut easily, therefore they are hydrolyzed by intestinal enzymes before entering the colon [101]. Wang et al. [107] studied how gut microbiota participated in the biotransformation of flavonoids in the giant panda. Intestinal enzymes, namely, α-L-rhamnosidase and β-glucosidase are involved in the conversion of flavonoids and α-glucosidase also might contribute to the utilization of flavonoids, which plays a crucial role in glucose metabolism and pharmacological action. Gut microbial enzymes associated with metabolism of flavonoids are detected in the gut microbiota. Enzymes with high sequence similarity can catalyze distinct chemical reactions. Different flavonoids are metabolized by different enzymes that are shown in Table 3.

During the absorption, flavonoids are firstly hydrolyzed in the intestine, prior to C-ring reduction and hydrogenation, and then transformed into phenolic acid by absorption or excretion. This process mainly includes O- and C-deglycosylation, demethylation, dehydroxylation, ester cleavage, reduction of carbon-carbon double bonds, isomerization, ring fission, extension and truncation of the aliphatic carbon chain and decarboxylation, which would play a significant role in improving the bioavailability of flavonoids as well as enhancing the activity of metabolites. However, changes in intestinal flora would affect the conversion of flavonoids, and in turn, flavonoids play a part in modulating the composition and richness of intestinal flora. Therefore, metabolomics, intestinal metagenomics and microbial culture techniques are used to explore the interaction between flavonoids and gut microbiota [119]. An ultra-high-performance liquid chromatography with LTQ-Orbitrap mass spectrometer (UHPLC-LTQ-Orbitrap) method is often used to identify the metabolites in urine, plasma and feces samples in animals. Jiao et al. [120] elucidated the metabolism in rats after oral administration of hesperidin (HPD) and hesperetin (HPT). In general, the metabolic pathway is a combination of several reaction types. Diglucuronide conjugation was found in rats after oral administration of HPT, as hydrolysis, demethoxylation, dehydration, dehydrogenation, demethylation, glucuronide conjugation, sulfate conjugation and N-acetylcysteine conjugation after oral administration of HPD. Here are several common metabolic reactions summarized as follows:

**Deglycosylation reaction.** Flavonoids are mostly bound to sugars in plants to form glycosides, which are converted to glycosides via α-rhamnosidase and β-glucosidase enzymes produced by intestinal microbiota. The gut microbiota is mainly distributed in *Bifidobacteriaceae*, *Lactobacillaceae*, *Lachnospiraceae* and *Enterococcaceae* [104]. The first step of metabolism is the deglycosylation of flavonoid glycosides, which occurs in the intestinal lumen, inside or outside enterocytes. Cytoplasmic β-glucosidases present in the small intestine and liver hydrolyze various phenolic glucosides, which have a high affinity for quercetin glucosides and are inclined to be absorbed by small intestinal epithelial cells. Chlorogenic acid is neither hydrolyzed by epithelial cell cytoplasmic enzymes nor absorbed [121].

**Demethylation and dehydroxylation reactions.** As usual, the phenolic A- and B-rings of flavonoids carry hydroxyl and/or methoxy groups. Flavonoids would be metabolized by enzymes after absorption with demethylation and dehydroxylation occurring [122]. *Eggerthella lenta, Adlercreutzia equolifaciens* and *Escherichia* strain can, respectively catalyze the dehroxylation of flavan-3-ols and flavone [123].

**Deketomethylation reaction.** Isoflavone glycosides are converted via deketogenization to equol by gut microbial enzymes [124]. It was proved that polymethoxyflavones (PMF)-metabolizing bacterium *Blautia* spp. MRG-PMF1 could convert formononetin into daidzein by demethylation [125].

**Reduction and C-ring cleavage reactions**. Under the anaerobic gastrointestinal tract conditions, the double bond between C2 and C3 of flavonoid glycosides is reduced by hydrogenation. Reductive cleavage would also occur between the bonds O2 and C2, resulting in the opening of C-ring. Flavonols are passively diffused to the colon wall through degradation into phenolic acids by C-ring cleavage of flavonoids. *Eubacterium oxidoreducens*, *E. ramulus*, *E. casseliflavus*, and *Clostridium orbiscidens* that belong to *Butyrivibrio* genus are involved in the C-ring breakdown [126]. With the effect of *Eubacterium ramulus* Julong 601, genistein was converted into 2-(4-Hydroxyphenyl) propionic acid via C-ring cleavage [127]. *Clostridium* and *Eubacterium desmolans* are desmolytic for the C-17 to C-20 bond of 17-hydroxycorticoids and cleave the C ring of naringenin in yields of 20 to 25% [128].

The biotransformation of flavonoids by gut microbes is of great significance to maintain the human health. Some studies have shown that most flavonoids are degraded to related phenolic compounds by C-ring cleavage [128]. *Bifidobacterium* and *Lactobacillus* strains could metabolize flavonoids such as soy glycosides and quercetin into esmolol and phenolic acid, respectively by Lipopolysaccharide (LPS)-stimulated peripheral blood individual nucleated cells with anti-inflammatory activity. Meanwhile, flavonoids exert the probiotic properties on beneficial intestinal bacteria [33]. The metabolism of flavonoids by intestinal microorganisms provides a basis for elucidating in vivo metabolic processes of flavonoids.

### 4.3. The Role of Gut Microbiota in Dictating Bioavailability and Bioactivity of Flavonoids

Bioavailability is defined as the degree of digestion, absorption, metabolism, and excretion of a kind of compound after ingestion. LC-MS/MS is used to measure plasma concentrations and/or urinary excretion among adults after ingesting a single dose of flavonoids to investigate the kinetics and extent of flavonoid absorption [129]. An in vitro cell co-culture model is used to simulate the physiological environment of the small intestine in vivo and further study the metabolic transport of drugs in the gastrointestinal tract. Human intestinal Caco-2 cells are used for estimating the bioavailability of flavonoids because they could simulate the absorption and transport process of small intestinal epithelial cells. Bioavailability is a crucial factor that determines their biological activity in vivo.

The in vitro co-culture model of Caco-2 intestinal cells and intestinal floras is universally used to assess the harm risk caused by the bacteria to human health. The 16S rDNA fragment of intestinal flora is quantified by quantitative fluorescence PCR. Trans-epithetical resistance (TEER) and the immunofluorescence staining image of tight junction protein are used to assess the integrity of Caco-2 intestinal epithelial cells. Bian et al. [130] studied that kaempferol exerted the potential effect on LPS-induced inflammation and barrier dysfunction in a coculture model of intestinal epithelial cells and intestinal microvascular endothelial cells. Kaempferol alleviated the drop of TEER and decreased the protein expression of zonula occludens-1 (ZO-1), occluding, and claudin-2. The barrier dysfunction of the caco-2 monolayer in the LPS-induced epithelial-endothelial co-culture model was alleviated by the kaempferol.

Flavonoids are generally ingested orally, while intestinal microorganisms participate in various metabolic reactions, most of which are in the large intestine. The catalytic action of microorganisms includes hydrolysis, cleavage and reduction, which are further divided into demethylation and dehydroxylation. Flavonoids exist mainly in the form of glycosides, which must be removed before absorption and cleaved into different products by microbial enzymes. Intestinal absorption is affected by many factors, such as the structure of flavonoids, the types of metabolic enzymes, the expression and function of intestinal transporters and metabolic enzymes, which will affect the stability, solubility and permeability of flavonoids and eventually lead to low oral utilization, thus affecting the efficacy and safety [131]. Methylated flavonoids have improved bioavailability that helps understand the structure and bioactivity [132]. In addition, the phospholipid complex is used to enhance oral bioavailability of flavonoids. The metabolic processes mediated by the liver (phase I and II metabolism) as well as the intestine and microbiota have an important influence on the bioavailability of flavonoids [133].

There are many factors affecting bioavailability, such as the chemical structure, sugar binding and metabolic enzymes of flavonoids, plasma protein binding and the participation of intestinal microorganisms in various metabolic reactions. The chemical structure of flavonoids restricts the absorbance of bioactive and biotransformation. It is proved that the type of initial glycosylation and position of the hydroxyl groups affect flavonoids degradation rate. The richness and diversity of gut microbiota may influence the biotransformation and metabolites produced from ingested flavonoids because specific gut microbial species and genera with special gene encode precise enzymes for specific chemical reactions [126]. *Bifidobacteria* and *Lactobacillus* are potential probiotics to promote the production of bioavailable catabolites. In the transformation process of flavonoids, microbial catalysis includes hydrolysis, cleavage and reduction, which can be further divided into demethylation and dehydroxylation. Flavonoids exist mainly in the form of glycosides, which must be removed before absorption. Through the action of microbial enzymes, the flavonoids bioavailability improves. Bioactive flavonoids usually undergo phase I and II metabolism. Phase II metabolic reactions form hydrophilic compounds that favor the deconjugation reactions. The presystem activity is important to enter the next enteral and entero-hepatic recycling [131]. Most of flavonoids are not absorbed in the enterohepatic circulation, but instead metabolized by gut microbiota. Thus, metabolites produced by gut microbiota may play a crucial role in auxiliary health benefits [134]. The chemical forms of metabolites are deserved to be studied in systemic circulation to understand the action mechanism of flavonoids. It is necessary to study the interaction between gut microbiota and metabolites in detail for further development of microbiota-flavonoid interaction [135].

There exist individual discrepancies in hesperidin bioavailability in human trials and animal studies. The variability of hesperidin bioavailability may correlate with enzyme activity and gut microbiota. Hesperidin is metabolized in the colon by α-rhamnosidase and β-glucosidase activity of gut microbiota, then released into the blood in the form of glucuronic acid and sulfuric acid combination and distributed systemically to the whole body, thereby influencing the metabolome and regulating the host physiological function. Metabolites of hesperidin such as 3-(3′-hydroxy-4′-methoxyphenyl) propionic acid (HMPPA) were transformed by colon intestinal microbiota that resulted in the increased bioavailability. The clinical studies indicate the bioavailability of hesperidin is associated with the content of hesperidin excretors by gut microbiota enzymes [136]. Flavonoids are metabolized by intestinal microorganisms to decompose into secondary metabolites, which are absorbed into the human body. It is detected in feces with high bioavailability, indicating that it has good biological activity and possesses efficacy on the regulation of diseases, which provides a guideline for preventing diseases. Although many studies have proved that the beneficial health effects are attributed to their antioxidative and biological properties, the actual bioavailability of flavonoids is still unclear.

Due to the inefficient systemic delivery and poor oral bioavailability of flavonoids, food macromolecule-based nanoparticles can be applied in the manufacture of food proteins, cross-linked polysaccharides, proteoglycan conjugates, and emulsified lipids. It plays a role in improving the bioavailability of flavonoids in blood via enhancing their solubility, but how the stability of nanoparticles and their structure affect the metabolism of flavonoids needs more focused attention in further studies [137]. Encapsulation of polyphenols in different matrices improves polyphenol release and subsequent absorption to exert target-specific health effects, and some studies have shown the beneficial effect of encapsulation on improving bioavailability and bioaccessibility [138]. Nanoparticle presentation, microencapsulation and fermentation are the prevailing methods of improving bioavailability [139].

## 5. Effects of Flavonoids on Intestinal Microbiota

Interaction between bacteria and their host can be recognized as reciprocal integrity between symbiosis, commensalism and pathogenicity [140]. Function components such as unabsorbed dietary flavonoids and their metabolites may play a significant role in the intestinal environment by modulating the microbial richness, diversity, and composition of the intestinal microbiota (Table 4). Flavonoids exert the prebiotic-like effect on stimulating the growth of beneficial microorganisms as well as reducing harmful bacteria, thus targeting the regulation of intestinal bacteria. The effects of flavonoids on intestinal microbiota are mainly reflected in four aspects: (1) modulation of the composition and richness of intestinal flora; (2) protection of intestinal barrier function; (3) influence on the immune system; and (4) influence on the gut microbiota metabolites.

### 5.1. Modulation of the Composition and Richness of Intestinal Flora

After entering the intestinal tract, flavonoids, which are decomposed by intestinal microbiota, regulate the composition and abundance of the intestinal flora, thereby affecting the physiological functions and playing a role in preventing and treating diseases. Specifically, on the one hand, flavonoids play a role in promoting the growth of the relative abundance of beneficial flora, and the inhibition of pathogenic bacteria, thereby affecting the richness and composition of intestinal flora reproduction [154]. Flavonoids inhibit the formation of bacterial biofilm and the energy metabolism of bacteria, thus improving the ecological balance of the intestinal tract. On the other hand, flavonoids have an impact on the enzyme system of intestinal microorganisms mainly through several paths such as affecting the types and number of enzymes in the intestine and chelating with metal ions to form insoluble compounds, resulting in dysfunctionality due to the lack of cofactor.

There is a certain correlation between the content of flavonoids absorbed in the human body and the diversity of gut microbiota. Here, taking advantage of metabolomics and macro genomics, it is found that the more the flavonoids contain, the lower the diversity of microbiome is. The experiment reported by Wang et al. [107] screened flavonoids in feed, excrement, urine and plasma. The low detection rate of flavonoids found in blood plasma by liquid chromatography-mass spectrometry (LC–MS) showed that most flavonoids could not enter the circulation of blood directly but were decomposed in the intestine, thereby producing a systematic effect in the body. Other metabolic or newly generated monomer flavonoid compounds by gut microbes were widely utilized and biologically transformed, with gut microbes exerting local effects in the gut. Metagenomes were characterized in real-time by shotgun sequencing to explore microbial pathways that may be involved in flavonoid transformation. Fecal bacteria culture in vitro with flavonoids provided the effect of flavonoids on intestinal microbial composition. At the omics level, flavonoids had an inhibitory effect on *Bacillus* (OTU9) and *Enterococcus* (OTU35). The combined analysis also showed that CAG00082 (*Bacillus*) was negatively correlated with flavonoids.

Different flavonoids have different impacts on bacteria, and modulation on gut microbiota contributes to the treatment and prevention of diseases. While certain bacteria groups are inhibited, some beneficial bacteria can thrive in the gut ecosystem [155]. Flavonoids have been linked to an increase in *Bifidobacterium* and *Lactobacillus*, which is associated with the reduction of obesity incidence and a decrease in *Staphylococcus aureus*, *Salmonella typhimurium* as well as pathogenic *Clostridium* species, which is beneficial to a decrease in the incidence of IBD disease [156]. Luteolin has a significant impact on the regulation of intestinal microbiota. Luteolin can change the structure of the gut microbiota by hierarchical cluster analysis in an animal model [157]. Porras et al. [158] showed that quercetin, a flavonoid with antioxidant and anti-inflammatory properties, could diminish hepatic lipid accumulation and improve insulin resistance as well as regulate lipogenic genes through modulating the gut microbiota. Due to the role of gut microbiota, the prebiotic capacity of quercetin and the transfer of established metabolic profiles provide a scientific basis as a protective strategy to prevent obesity-related non-alcoholic fatty liver disease (NAFLD). It also has shown that flavonoids isolated from cyclocarya paliurus are beneficial to maintaining microecological balance by modulating the flora structure with the proliferation of *Bifidobacterium* and *Lactobacillus* [159]. Cassidy et al. [160] found that feeding high and low doses of hesperidin in healthy mice showed a great enrichment of bacterial synthesis in the cecum, promoting an increase in the proportion of *Staphylococci*, with an additional increase in *Lactobacillus* levels in the high-dose group. Larkin et al. [8] found that soy isoflavones interacted with colonic flora to produce novel microbial transformers with bioactivity, which improved the bioactivity and bioavailability of soy isoflavones. Xu et al. [161] showed that glucoraphanin from broccoli seeds had an effect on regulating intestinal flora accompanied by an increasing intestinal flora diversity as well as restoring intestinal flora composition by 16S rDNA sequencing in high-fat diet-induced obese mice, thus it is a great potential to treat obesity associated with the modulation of intestinal flora dysbiosis.

Given that the beneficial effects of flavonoids are mostly remodeling gut microbiota, several bacteria demonstrate the changes in regulating human health. The relative abundance ratio of *Firmicutes* and *Bacteroidetes* (*F/B*) is a crude indicator of bacterial movement, both of which are dominant in the human intestine. Extracts of citrus flavonoids were shown to ameliorate diet-induced obesity and metabolic syndromes such as NAFLD, dyslipidemia, and insulin resistance with metabolic protection mechanisms related to gut microbial regulation by lowering *F/B* [162]. In addition, *Akkermansia* and *Allobaculum* are linked to obesity-related metabolic syndromes which play a protective role in the gut. An increase in the relative abundance of *Akkermansia* and *Allobaculum* was identified in high-fat diet-induced obesity mice, while weight loss intervention in flavonoid-rich citrus peel extracts was accompanied by a higher abundance of both. Bekiares et al. [163] studied that dietary flavonoid metabolism also promoted the production of *Akkermansia* spp. Recent studies have shown that the muciniphilin-like bacteria *Akkermansia muciniphila* along with *Enterococcus faecalis prausnitzii* and *Ruminococcus* are considered as the next generation of health-promoting intestinal bacteria [164]. Turnip sulfur, a kind of flavonoid that acts as a functional component in the regulation of lipid metabolism, has been reported to become a potential anti-obesity substance and is associated with weight loss in a mice weight loss intervention. Turnip sulfur incubation led to a reduction of *Firmicutes* to *Bacteroidetes* ratio and the abundance of *Clostridium* by assessment of the high-fat diet rice bacteria which was reported to be negatively correlated with diabetes and obesity. Intervention with turnip sulfur may protect the activity of antioxidant enzymes from oxidative stress induced by a prolonged high-fat diet [165]. In summary, flavonoids have modulatory effects on related diseases by affecting the proliferation of intestinal microbiota.

### 5.2. Protection of Intestinal Barrier Function

The intestinal mucosal barrier is a protective barrier system in the intestine that functions as a biophysical barrier that separates microbe-enriched luminal environment from the host, which prevents harmful bacteria, antigens and therapeutic substances from entering the cells and prevents bacteria and endotoxins from moving in the intestinal cavity, to achieve gut health. It is maintained by tight junctions, adhering junctions and desmosomes. There is an association between intestinal barrier disorders and microbial dysbiosis. The microbial dysbiosis will cause translocation of lipopolysaccharide and damage to the structure and function of tight junction. A functional intestinal mucosal barrier includes three components, mechanical barrier, immunological barrier and ecological barrier, which interacts with each other to exert influence [166].

Intestinal tract interacts with microbiota, which arouses the stimulation towards intestinal epithelial cells. Interaction between the intestinal barrier and gut microbiota exists. On the one hand, a functional intestinal barrier can regulate the richness and composition of bacterial species. In turn, the gut microbiota can regulate the intestinal barrier integrity and repair the intestinal mucosa by modulating innate and adaptive immunities [167]. As a rule, gut microbiota plays an indispensable role in maintaining the function of the intestinal barrier. However, when pathogenic bacteria increase owing to various reasons, the intestinal environment would be easily destroyed. Moreover, the intestinal microbiota is abundant in species, some of which are beneficial bacteria exerting different effects such as anti-viral, anti-bacterial, anti-inflammatory and enhanced immunosuppressive anti-tumor. Wang et al. [168] showed that flavonoids had health-care effects on intestinal barrier function, which was involved in the protection of barrier permeability, positive regulation of intestinal microbiota, modulation of intestinal immune system, as well as inhibition of oxidative stress and inflammation in the lumen of the intestine such as diet-induced obesity. Therefore, flavonoids may be effective candidates for the prevention and treatment of diseases associated with barrier dysfunction.

Flavonoids are absorbed and metabolized by the intestinal tract and are transformed into metabolites on epithelial cells to act on the intestinal barrier and regulate the infiltration of the intestinal mucosa and the immune system [168]. Flavonoids have a health-promoting effect on the intestinal barrier function through the following mechanisms: (1) inhibition of inflammatory signaling; (2) protection of barrier permeability; (3) intestinal lumen oxidative stress; (4) immune regulation; (5) up-regulation of GLP-2 improves the advocacy barrier function; (6) regulation of the intestinal tight junction barrier and structural integrity; and (7) active modification and remodeling of the intestinal microbiota [168,169].

Phytochemicals can ameliorate mucosal inflammation that triggers barrier dysfunction. Impairment of ZO-1 expression induced by TNF-α and the decrease in the expression of nuclear factor kappa B (NF-κB)-dependent inflammatory cytokines (interferon (IFN)-γ, interleukin (IL)-1β, IL-17) may induce inflammation and lower barrier function. Phytochemicals improve gut barrier function via the modulation of intestinal immunometabolic regulators. Phytotherapies improve the tight junction protein expressions such as claudin-1, occludin, ZO-1, junctional adhesion molecule A (JAMA), and epithelial mucin barrier and induce intestinal secreted immunoglobulin A (sIgA) [170]. Highbush blueberries contain anthocyanins and proanthocyanidins that influence gut barrier dysfunction. Polewski et al. [171] demonstrated that anthocyanin-enriched fractions were able to attenuate the gut permeability and subsequent gut barrier dysfunction [172]. It protected the intestinal epithelium against different luminal challenges, thus helping to promote intestinal health and mitigate chronic intestinal and metabolic diseases. Sharma et al. [173] showed that flavonoids reduced oxidative stress and membrane leakage, further reduced paracellular permeability and remodeled the TJ-associated proteins in hyperglycemia-induced Caco-2 cells, to enhance intestinal barrier functions.

Although the effects of flavonoids on intestinal microbiota have gained much attention in recent years, there are still some limitations when studying the interaction between flavonoids and intestinal microbiota: (1) The regulation of flavonoids in the intestinal flora usually manifests the promotion or inhibition of gut microbiome, but the effect on specific intestinal flora is not clear. (2) Flavonoids extracted by plants have been analyzed without considering the bioavailability and metabolic mechanism in individual compounds. (3) When studying the role of flavonoids in gut microbiota, both human and animal experimental doses are high, which cannot represent the regular diet, and it is less significant to practical guidance. There are still insufficient studies observing in vivo impact of flavonoids on gut microbiota. (4) There are few studies on the in vivo effects of flavonoids on human intestinal microbiota and individual differences that have an impact on their interaction should be taken into consideration [174].

### 5.3. Influence on Mucosal Immunity System

Intestinal microbes and host immune system are in coadjustment. The gut microbiota has a highly coevolutionary relationship with the host immune system. Gastrointestinal microbiota involves in intestinal epithelial morphology and immune system function [175]. The intestinal mucosa is protected by the immune system. The innate mucosal immunity includes physicochemical barriers formed by the gastrointestinal epithelium (digestive enzymes, mucin, peristalsis, etc.) to protect the host from the influence of intestinal microorganisms. Intestinal composition and metabolites not only but also regulate metabolic system, but also promote the maturation of the host immune system by providing the beneficial metabolites. Immune system plays a crucial role in maintaining gut homeostasis [176]. SCFAs could act as the signaling molecules that influence the immune system. A study has shown that gut microbiota has promoted dozens of metabolites to exert systemic effects. A metabolic pathway that gut symbiont *Clostridium sporogenes* generates aromatic amino acid metabolites was found by genetics and metabolic profiling. It is proven that gut microbiota affects intestinal permeability and systemic immunity [177].

Flavonoids up-regulate the production of beneficial bacteria, which can stimulate the secretion of secretory immunoglobulin to prevent the adhesion of pathogenic bacteria and toxins, inhibit the imbalance of other microorganisms, and maintain the balance of intestinal flora through the placeholder effect and the secretion of antimicrobial peptides. By constructing a biological barrier, it promotes mucin expression and secretion, enhances intestinal mucosal barrier function, prevents harmful bacteria and endotoxin from migrating, and regulates intestinal health. Flavonoids enable to remove and alter host cells and change specific immune responses. For adaptive immunity, Th cells can reduce the pro-inflammatory process and seem to prevent autoimmune gastritis and inflammatory bowel disease [178].

### 5.4. Influence on the Gut Microbiota Metabolites

Human intestinal flora produces bile acids, short-chain fatty acids, ammonia, phenols, endotoxins and other bioactive compounds under the stimulation of dietary nutrients. These microbial-derived metabolites are the core between the microorganism and the host that provide further insight into the influence of lifestyle and dietary factors on disease [179]. Interaction between gut microbiota and metabolites helps understand how flavonoids influence human health.

Short-chain fatty acids are the main products of intestinal microbial fermentation. These short-chain fatty acids are potential targets for the treatment of diseases. SCFAs have functional impacts on the host. SCFAs are the regulators of gut metabolism, proliferation and differentiation. For example, SCFAs can promote the secretion of glucagon-like peptide (GLP)-1 and peptide YY (PYY) in enteroendocrine cells through the peptide GPCRs GPR43 (FFAR2) and GPR41 (FFAR3) [180]. In addition, SCFAs participate in the barrier function and immune responses. The important role in maintaining a favorable environment for commensal bacteria and controlling pathogens’ growth has attracted more attention [181]. Baicalin inhibits Avian pathogenic *Escherichia coli* (APEC)-induced lung injury by regulating gut microbiota and SCFA production. The production of SCFAs including acetic acid, propionic acid and butyric acid, acetic acid is increased. In addition, the concentrations of acetic acid and its receptor free fatty acid receptor 2 (FFAR2) were significantly upregulated to treat APEC infection [182].

### 5.5. Affecting the Expression of Signaling Molecules

Epigenetic modifications are important for transcriptions, antibody maturation, DNA methylation and chromosome stability [183]. Dietary phytonutrients can shape gut microbes and intestinal mucosa via epigenetic modifications. A study has found that flavonoids shape gut microbiota and further regulate intestinal mucosa functions via affecting epigenetic modifications of gut tissue, modulating intracellular receptors and signaling molecules. It also shed light on manipulation of gut microbiota by dietary nutrients, for prevention or clinical treatment of intestinal diseases [184]. Many chronic diseases are caused by microbiota-induced intestinal dysfunction. Thus, the maturity and the relative stability of gut microbiota are vital for gut health. Flavonoids are shown to achieve the proper balance between gut microbiota and health of the host.

Gut microbiota can mediate epigenetic modifications in the gut via direct or indirect mechanisms, and flavonoids affect epigenetic enzymes via epigenetic regulation. Types of epigenetic modifications include DNA, RNA, methylation, β-hydroxybutyrylation, formylation, crotonylation, butyrylation, etc [185]. Gut microbiota can produce metabolites such as SCFAs regulated by flavonoids, which play an important role in the induction of acylation modifications and are critical signaling molecules for maintaining epithelial health and gut health. Microbial colonization regulates global histone and acetylation and methylation in host tissue in a diet-dependent manner [186]. The study has shown that flavonoids can induce the expression of different tumor suppressor genes that may contribute to decreasing breast cancer progression and metastasis [184]. Wu et al. [187] found that luteolin inhibited the proliferation and metastasis of androgen receptor-positive triple negative breast cancer (TNBC) by epigenetic regulation of MMP9 expression via the AKT/mTOR signaling pathway.

## 6. Flavonoids, Gut Microbiota and Health

Research on the regulative function of gut microbiota has broadened our cognition of human diseases and its underlying mechanisms. From the outcome of recent epidemiological, physiological and omics-based studies, complemented by cellular studies and experiments in animals, more studies on the health effects of interactions between flavonoids and gut microbes should be valued due to the connection with diseases. Overall, during the metabolic process, flavonoids can be hydrolyzed into glycosides, glucuronides, sulfates, amides, esters and lactones by gut microbiota via various biochemical reactions. Many beneficial or harmful metabolites such as lipopolysaccharides, peptidoglycans, trimethylamines, secondary bile acids, and SCFAs are produced to regulate the diseases [188]. On the one hand, the interaction between flavonoids and intestinal microbiota regulates the composition of gut microbiota in the gastrointestinal tract channels and the functional enhancement of probiotics. On the other hand, the bioavailability and bioactivity of flavonoids are significantly improved, which exert a regulatory effect on intestinal diseases. It is high time that the correlation between flavonoids and gut microbial function should be unraveled to shed light on how to regulate diseases. Accordingly, some typical diseases are involved in the research hotspot to reveal the correlation between flavonoids and gut microbial function.

### 6.1. Regulation of Flavonoids on Gastrointestinal Diseases

The potential of flavonoids for gastrointestinal health and as a therapeutic agent in gastrointestinal health is attributed to its effects on influencing the intestinal barrier, intestinal immune system, nutrients digestion and absorption, and microbiota growth and metabolism. The main mechanisms involve maintaining the integrity of the intestinal mucosa as well as activating signaling pathways associated with the intestinal mucosal barrier of intestinal epithelial cells for preventing pathogen invasion. Chen et al. [189] studied that baicalin has the capacity of protecting intestinal epithelioid cell 6 (IEC-6) cells and intercellular tight junctions from LPS-induced injury by inhibiting the production of inflammatory cytokines.

Inflammation is considered to be the most common factor to induce gastrointestinal disease, which begins with interrupting the inflammatory cycle of reactive oxygen species and stimulating the production of antioxidant enzymes thereby reducing oxidative stress. It is the most obvious immune defense, which is manifested by swelling, pain, heat and redness in the affected tissue and is characterized by increased cytokine production such as IL-1β and TNF-α to stimulate inducible nitric oxide synthase (iNOS) and nitric oxide (NO) production to increase the generation of reactive oxygen species (ROS) [190]. The previous study demonstrated that almost all flavonoids had varying degrees of anti-inflammatory properties [191]. Evidence suggested that naringenin exhibited anti-neuroinflammatory action by inhibiting iNOS expression, NO production, p38 phosphorylation, downstream signal transducer and activator of transcription-1 (STAT-1) in neuronal glial cultures [192]. More in vivo and in vitro studies have focused on the anti-inflammatory activities in regard to flavonoids such as hesperetin and hesperidin. Flavonoids can inhibit the inflammatory process mediated via glial cells by accessing the brain [193]. Hosseinzadeh et al. [194] demonstrated that quercetin could inhibit the inflammatory signal TNF-α-dependent, NF-κB activation to prevent intestinal tumor recurrence. Moreover, rutin could be deglycosylated to quercetin to decrease the damage of the colon in acute and chronic colitis induced by trinitrobenzene sulfonic acid (TNBS) in rats and act as the anti-inflammatory effect. In vivo and in vitro evidence supported the fact that dietary flavonoids reshaped systemic by reducing intestinal permeability and improving local inflammatory conditions to metabolic homeostasis and thus prevented systemic inflammation from maintaining the metabolic stability of the body in non-intestinal models [195].

Meanwhile, it is of vital importance to maintain the specific immune capacity of the intestine as well as improve intestinal-specific sensitivity of the immune response, as relevant inflammatory signaling pathways were inhibited, thereby quickly reducing the production of pro-inflammatory factors and protecting intestinal epithelial cells [196].

IBD is a group of chronic recurrent intestinal inflammatory diseases characterized by the relapse of immune system and inflammation, whose clinical reactions are featured by repeated abdominal pain, diarrhea, mucus and bloody stools, and even a variety of systemic complications such as blurred vision, joint pain, rash, etc. Ulcerative colitis, a clinically common inflammatory bowel disease, got improved by luteolin, which contains lignin that is capable of reducing injury in rats. In UC rats, the decreasing NF-κB, IL-17 and IL-23 levels and the increasing peroxisome proliferator-activated receptor (PPAR)-γ levels accounted for the inhibition of colonic inflammation [148].

Flavonoid natural products exert antioxidant effects on reducing oxidative stress and inhibiting the activation of various signaling pathways to improve inflammatory bowel disease [197]. Flavonols have the capacity of affecting the composition of intestinal flora with the growth of beneficial bacteria such as *Lactobacillus* and *Bifidobacterium*, while reducing the composition of *Clostridium spp.* and improving the dysregulation of intestinal homeostasis *in vivo*. Pei et al. [198] showed that Cynanchun thsioides (Freyn) K. Schun is a promising herb for the treatment of irritable bowel syndrome, rich in flavonoids, which has antibacterial and anti-inflammatory activity. Cynanchun thsioides was proved to adjust the intestinal microbial flora in IBS mice models, through the analysis of bacterial 16S rDNA gene, suggesting that it had a significant effect on reducing the abundance of *pseudomonas, Lachnospiracea_incertae_sedis* and *Clostridium XIVa*, and the reversal of visceral hypersensitivity, along with an increase in *Clostridium IV*, *Elusimicrobium*, *Clostridium_sensu_stricto*, and Acetatifactor. The latter four taxa played a significant role in regulating gastrointestinal tract function and manifesting a protective effect on the intestinal epithelial barrier by producing SCFAs or lithocholic acid (LCA) [199].

Lipopolysaccharide binding protein (LBP) is a polypeptide secreted into the bloodstream as a 58-60 kDa glycosylated protein. LBP mediates potent innate immune responses by recognizing LPS originating from different Gram-negative bacteria. It can monomerize LPS multimers that enable cellular recognition via CD14 and the toll-like receptor 4 (TLR4)-MD-2 receptor complex by promoting TLR4-dimerization. LBP can also detoxify LPS by transferring it to lipoproteins to limit inflammation. LBP can transfer LPS to membrane-bound or soluble CD14 and subsequently to the MD-2-TLR4 complex. It is the site of interaction with CD14 and mediates LPS transfer to MD-2-TLR4 [200]. Luteolin treatment resulted in an increase in the proportion of *Bacteroidetes* and a decrease in the proportions of *Firmicutes, Proteobacteria, Lactobacillus* and *Prevotella_9*, as well as the levels of NF-κB, IL -17 and IL-23, with an increase in PPAR-γ. In addition, luteolin treatment also enhanced the level of *Roseburia* and *Butyricicoccus* that could produce butyrate. Luteolin has an effect on NO and prostaglandin E2 (PGE2), as well as the expression of inducible iNOS, cyclooxygenase-2 (COX-2), TNF-α, and IL-6 in mouse alveolar macrophage MH-S and peripheral macrophage RAW 264.7 cells [201]. It also inhibits the expression of different clusters of differentiation (CD), particularly, CD-40, CD-80, CD-86 [202]. A study proved that extracellular signal-regulated kinases (ERK) inhibitor prevented the effects of luteolin, thus, luteolin is capable of activating the ERK signaling pathway to underlie the anti-inflammatory effect (Figure 2) [203].

### 6.2. Modulation of Flavonoids on Obesity and Diabetes

Obesity is caused by disturbance of energy balance that has a strong correlation with metabolic diseases. With the prevalence of obesity, the incidence of comorbidities such as T2DM, inflammatory bowel disease, cardiovascular diseases and cancer has increased, inflicting a high economic burden. More and more studies have shown that gut microbiota plays an important role in the pathophysiology of obesity, insulin resistance and metabolic syndrome. Intestinal population can directly regulate the expression of fat, energy metabolism, lipid metabolism and storage-related genes in the body, thus distorting capacity metabolism Excessive fat synthesis and lipid accumulation leads to the formation of obesity or other diseases. Intestinal flora can induce systemic inflammation and lead to obesity. For example, lipopolysaccharide produced by Gram-positive(G-) bacilli in intestinal flora is a prerequisite for obesity induced by a high-fat diet. LPS produced by G-bacilli binds to the complex receptor CD4 /TLR4 on the surface of immune cells to trigger the release of pro-inflammatory cytokines, causing an inflammatory response and metabolic disorders, thus inducing diseases such as obesity.

Links between obesity and gut microbiota have been discussed a lot [204]. Evidence suggested that dysbiosis may contribute to the onset and pathogenesis of obesity. Notably, the composition of gut microbiota is linked to obesity. In general, obese tends to present more *Firmicutes* and fewer *Bacteroidetes* than the lean [205]. In obese humans, lower diversity of the gut microbiota was associated with increased adiposity and dyslipidemia, impaired glucose homeostasis, and insulin resistance, caused by gut dysfunction. Gut microbiota influences the metabolism of glucose, whose metabolites could increase the release of insulin. Perry et al. [76] studied that gut microbiota-nutrient interaction in HFD-fed rodents contributed to the increased glucose-stimulated insulin secretion ghrelin secretion, obesity and its related sequelae of hyperlipidaemia, which is a possible therapeutic target for obesity.

In addition, the interaction between gut microbiota and flavonoids is related to such metabolic diseases as obesity and diabetes. Dietary intake appears to be a major regulator of the balance between obesity and gut microbiota. A high-fat diet may induce obesity by affecting microbiota and metabolites. Increasing evidence has shown that bioactive compounds such as flavonoids metabolites acting on intestinal microbiota can regulate the symptoms of obesity and diabetes by inhibiting lipogenesis and improving insulin resistance. Due to antioxidant and anti-inflammatory properties, flavonoids inhibit ROS synthesis, COX-2, and NF-κB signaling pathways and further exert an effect on obesity and inflammatory parameters [206]. Reportedly, it is proved that flavonoids have ameliorated HFD-induced gut dysbiosis by impacting microorganisms, with alteration in reduced weight and the modulation of lipid metabolism (Figure 3) [207]. In an obese db/db mice model, citrus flavonoids altered the diversity of gut microbiota and significantly attenuated plasma triglyceride and hepatic steatosis by altering inflammation and hepatic glucose- and lipid-regulating enzymes [208]. Yan et al. [209] demonstrated for the first time that *Anemarrhena asphodeloides*, one of the most commonly used herbs in traditional Chinese medicine that approved for diabetes treatment alleviates diabetes by regulating intestinal microbiota and protein expression. The inhibitory effect of *Anemarrhena asphodeloides* extract (AAE) on *Fasciola* spp., *Oligobacter* spp. and *Klebsiella* spp. facilitates the maintenance of intestinal microecological balance. AAE also promotes cell regeneration and restores islet cell function via Peroxiredoxin 4 (PRDX4) overexpression, which alleviates diabetes mellitus.

### 6.3. Impact of Flavonoids on Cancer or Tumor

Cancer is a major public health concern all over the world [210]. In recent years, the incidence of cancer is on a growing trend. The global cancer burden latest data in 2020 demonstrates that breast cancer ranked as the world’s largest with the rapid growth that replaced lung cancer. There is a large number of studies that demonstrates a complicated association between gut microbiota and cancer, including recent compelling evidence suggesting the role of gut microbiota-commensal *Bifidobacterium* in promoting antitumor immunity and facilitating anti-programmed death ligand 1 (PD-L1) efficacy, thus modulating cancer immunotherapy [211]. Gut microbiota, used as a model to explore the cancer mechanism, usually promotes carcinogenesis by disrupting microenvironmental homeostasis, exerting a systemic immunoregulatory effect, promoting inflammation, as well as secreting toxic metabolites. Increasingly, the gut microbiome is being recognized for its influence on diseases such as cancer as well. Generalized dysbiosis of gut microbiota may induce cancer under diverse influence mechanisms. Some bacterial species can potentiate intestinal carcinogenesis by inducing carcinogenic toxins to stimulate the production of reactive oxygen species, altering the immune response and activating oncogenic signaling pathways, thus the inflammatory state may cause the imbalance of intestinal flora [212]. Gut microbiota can influence bile acids and immunity system to further develop cancer [213]. Gut microbiota metabolites were produced by specific microbiota. For instance, *Clostridium, Bifidobacterium, Lactobacillus, Enterococcus, Faecalibacterium,* and *Roseburia* can hydrolyze secondary bile acids, thereby regulating secondary bile acid-mediated cell proliferating signaling [214]. TLR4 signaling is related to cancer by influencing the protease cathepsin K to promote tumor metastasis [213]. Gut microbiota is involved in the development of colorectal cancer. As per the experiment, probiotics were administered to colorectal cancer patients, and it was found that the bacteria producing butyric acid increased in the stool samples. Moreover, diet affects the composition of intestinal microbiota, which is often characterized by cross-omics and metabolomics studies. It is reported that flavonoid intake in some circumstances is involved in the cancer process and is capable of acting as a cancer-preventive agent [215].

Flavonoids have anti-tumor activity and antioxidant activity. They can regulate ROS and scavenge enzyme activities. More studies have shown that flavonoids can arrest cell cycle, inhibit the production of heat shock proteins and cancer cell proliferation. These all are relevant with the mechanism of preventing cancer [216,217,218]. Flavonoids affect the response of cells to oxidative stress, which is beneficial to the apoptosis of mutant cells. A recent study of luteolin in cancer proved promising results through the interaction of immune regulation and microbiota [219]. There is strong evidence showing that luteolin has great effects on inhibiting the growth of cancer cells in Classical Hodgkin’s Lymphoma via caspase activated-cell death, which is characterized by the reduction of C-X-C chemokine receptor type 4 (CXCR4), matrix metalloproteinase (MMP)-2 and MMP-9 [220]. The modulation of the STAT-3 pathway also influences the expression of micro-RNA in cancer. Kim et al. [221] studied that Ginkgo biloba leaf extract (GLE) suppressed human breast cancer resistance protein (BCRP) or multidrug resistance associated protein 2 (MRP2) expression in correlation with gut microbiota via Caco-2 cells in mice. Thus, the increase or decrease in certain gut microbiota may be correlated with in the intestinal BCRP or MRP2 expression levels. Another meta-analysis was conducted by Rienks et al. [222] to reveal the association between flavonoids and breast cancer through ten observational studies, showing that daidzein and genistein have a beneficial effect on preventing breast cancer risk with a certain concentration. The risk of cervical cancer was significantly increased in women with higher equol. Flavonoids mostly contain methoxy and hydroxyl derivatives. Baicalein, a major Scutellaria baicalensis, has historically been used for therapies, suggesting a potential therapeutic agent. Many studies have shown that baicalein has anti-cancer and anti-tumor activities against various cancer cells such as breast, gastric and so on. Different kinds of cancers have various targets and molecular pathways. An In vitro study showed that baicalein could inhibit proliferation and induce apoptosis in MCF-7 estrogen receptor positive (ER+) and MDA-MB-231 estrogen receptor negative (ER−) breast cancer cell [223]. Both in vivo and in vitro studies demonstrated that lipoxygenase (LOX) baicalein could inhibit the growth of pancreatic cancer cell and induce apoptosis through the mitochondrial pathway [224]. Moreover, baicalein showed chemopreventive and LOX-inhibitory activity by introducing a 12(S)-LOX expression vector into SW480 colorectal cancer cells to prove the model suitability [225]. Baicalein also inhibits migration and invasion of glioma cells by reducing cell motility and migration via the suppression of p38 signaling pathway [226]. Baicalein inhibits cancer cells generation, metastasis and inflammation via both mitochondrial-mediated and portable-mediated pathways, as well as the inhibition of cell cycle-dependent kinase (CDK) and cell cycle proteins B1, D1, and D3 in prostate cancer cells, lung squamous carcinoma [227], oral cancer and bladder cancer to induce cancer cell cycle arrest [228]. Gingko biloba leaf extract (GLE) contains bilobalide, ginkgolide, kaempferol, quercetin, flavonol glycoside. Pharmacokinetic studies showed GLE affected the intestinal transporter expression with the inhibition of drug transport by breast cancer resistance protein (BCRP) or multidrug resistance protein 2 (MRP2) and gut microbiota composition in mice. In vitro experience showed that GLE treatment decreased the populations of *Proteobacteria* and *Deferribacteres*. Facet metabolome profiles proved the involvement of gut microbiota in the modulation of BCRP expression.

Epidemiologic research results encourage that flavonoid intake could reduce the incidence of cancer, whereas high consumption of both flavonoids and isoflavone is associated with reduced risk of estrogen-related cancer [229] and in the European Prospective Investigation into Cancer and Nutrition (EPIC), the diet-cancer hypothesis was obtained from a large sample prospective cohort. In the Multiethnic Cohort (MEC) study, a wide range of data analyzed that a food pattern associated with the intake of flavonoids such as quercetin, kaempferol, as well as myricetin was linked with lower pancreatic cancer [230]. Diet-related cancer prevention recommendations were summarized in six points: body, body fat activity, foods and beverages that promote weight gain, animal-based foods as well as alcoholic beverages. Certainly, endogenous causes such as inherited mutations and immune conditions also would result in cancer. How flavonoids exert an effect on cancer can explain by cell cycle arrest, inhibiting epidermal growth factor receptor signaling pathways and activating DNA damage response pathway.

Cancer is becoming the most prevalent disease in the world, which could be treated by anti-cancer drugs, but different action targets and pathways of various kinds of cancers need to be further studied to understand the mechanism of how flavonoids accumulate in organelles and tissues in future studies. Flavonoids have the potential biological functions on regulating cancer, such as apoptosis, angiogenesis, cell differentiation, cell proliferation, etc. There is a correlation between the regulation of kinases induced by flavonoids and cell apoptosis, cell proliferation and tumor cell invasion behavior. Different targets and pathways deserve more studies to improve the disease and find out the interaction between flavonoids and key enzymes of tumor cells. Therefore, it is of great significance to explore how flavonoids regulate cancer and put forward new insights for fighting against cancer. It is necessary that future research starts with flavonoids to reduce side effects and whether painful chemotherapy could be substituted.

Understanding the bidirectional interactions between flavonoids and the gut microbiome and their impact on the clinical outcome of treatment for metabolic diseases may provide insights for the development of strategies to improve metabolic diseases in the next generation.

## 7. Conclusions

The interaction between flavonoids and intestinal microorganisms is reciprocal. After oral administration, only a small part of flavonoids can be digested and absorbed by the stomach and small intestine. After entering the colon, flavonoids are decomposed into glycosides through enzymatic decomposition and then fermented by anaerobic bacteria. It is beneficial to human health to decompose beneficial compounds into various microbiota metabolites. Furthermore, the metabolic capacity of intestinal microorganisms affects the absorption and function of flavonoids, thus affecting the bioavailability and bioactivity of flavonoids. In turn, the intake of flavonoids can regulate the composition and function of the gut microbiome. Intestinal bacteria are closely involved in regulating the immune system and promote the connection between the immune system and the gut by secreting substances, such as short-chain fatty acids. Short-chain fatty acids can regulate the intestinal barrier function by influencing immune system and maintaining tight connections, inducing anti-inflammatory molecules pathways, promoting the differentiation of T lymphocytes, improving adaptive immune cells, etc. Thus, it helps prevent and treat diseases. Therefore, understanding the bidirectional interaction between flavonoids and intestinal microbes and its impact on the clinical efficacy of various diseases may provide a guide for developing strategies to improve the next generation of metabolic diseases.

The substantial evidence supporting the potential for beneficial effects of flavonoids has led to more studies in this area and the development of new metabolic mechanisms between flavonoids and gut microbiota. However, the complexity of different diseases and the variability of gut microbiota mean that it is challenging to unravel the effects and shed light on the mechanisms of the interaction between flavonoids and intestinal microbes. It is difficult to clearly explain the relationship between the metabolites of flavonoids biotransformed and the biological activity of their parent compounds, and the contribution of metabolites remains unclear. This review has focused on the health-beneficial effects of flavonoids via gut microbiota, which is beneficial to further promoting the mechanisms of physiological function and provides an important theoretical basis for the application of flavonoids in regulating intestinal microbes and preventing related diseases induced by intestinal flora disorder.

## 8. Future Perspectives

In future research, metabolomics combined with microbiome research to study the interaction between flavonoids and intestinal bacteria is a new trend. It is a necessity to establish in vitro intestinal bacteria to develop and interact with intestinal bacteria of flavonoids in the database, further promoting the development of oral flavonoids drugs. Various new technologies can be used, such as chemical modification, enzymatic modification and nano preparation to strengthen in vivo analysis of flavonoids, which is conducive to the development and utilization of flavonoids. Future research directions are possibly in-depth into the potential mechanisms of flavonoids, including the identification of related molecular targets and signaling pathways, and the possible differential effects on the ultimate health effects. Additionally, further understanding of the underlying impact of diverse specific flavonoid classes, various doses and regiments on a various range of diseases is needed by combining and correlating microbial community. In addition, the impact of ethics, age, health and prevalent diet variations is needed to be taken into consideration. In this case, according to such assessment, prebiotics and probiotics can be used to understand the health benefits of flavonoids on a broader scale.

These opinions about the future perspective can provide a guide as follows:(1)For specific flavonoids, the effects of different doses on flavonoids need to be studied to determine the main factors associated with the overall association.(2)The effects of flavonoid intake on intestinal flora and whether it is synergistic or antagonistic in the treatment of diseases can be discussed in the future research.(3)Target flavonoid delivery methods, include protein peptides, monoclonal antibodies, and living cells, to study the effects of flavonoids on the microflora.(4)More studies could combine with clinical, cellular, and animal analysis of the safety and efficacy of IBD.(5)The function of microbial metabolites through multi-omics methods, including metagenomic, metaproteomic and metabolomic deserves to be studied for characterizing the metabolic pathways of small molecules produced by microbial metabolites.(6)The complex interaction between microbial host and flavonoids also needs to be further explored.(7)The dose of each flavonoid should be studied to understand the bioavailability of flavonoids, standard extraction and purification methods.(8)Various pathways have effects on diseases, and it is worth investigating whether these pathways are synergistic or cause side effects.(9)Effects of treatments on enzymes can be studied to evaluate the anti-disease potential of clinical experiments to clinical use of flavonoids.(10)By associating the metabolism of flavonoids with organisms, genes and enzymes, the investigation of human intestinal isolates will be conducted to identify individual organisms with specific metabolic abilities. The metabolite action will be detected by means of stable isotope detection, fluorescence in situ hybridization or imaging mass spectrometry.

## Figures and Tables

**Figure 1 foods-12-00320-f001:**
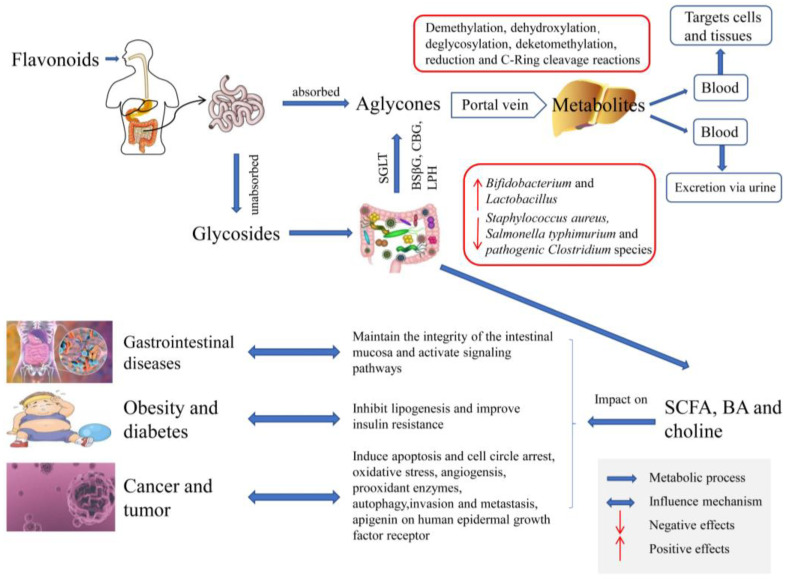
The metabolic mechanisms of flavonoids by gut microbiota and impacts on health. SGLT: Sodium-dependent glucose transporters; BSβG: Broad-specific β-glucosidase; CBG: Cytosolic β-glucosidase; LPH: Lactosephlorizin hydrolase; SCFAs: Short-chain fatty acids; BA: Bile acids.

**Figure 2 foods-12-00320-f002:**
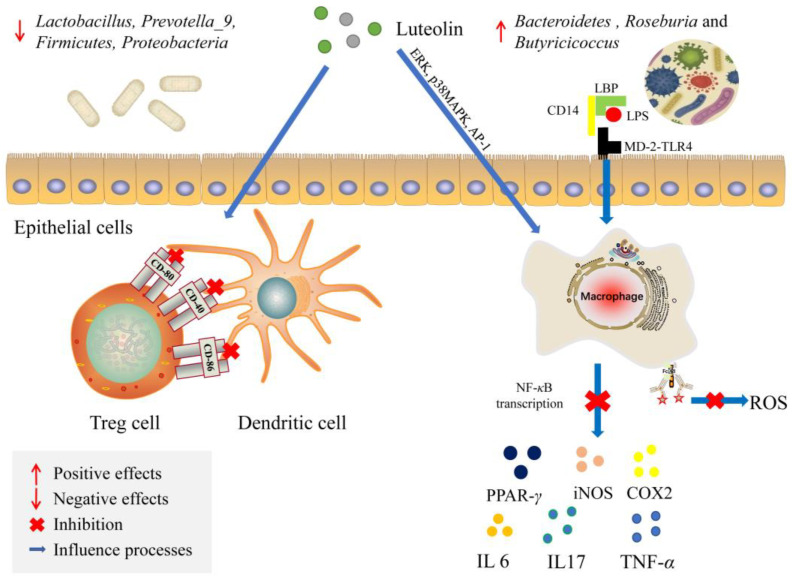
Regulation mechanisms of luteolin in the inflammation. ROS: Reactive oxygen species; LPS: Lipopolysaccharide; LBP: Lipopolysaccharide binding protein; CD: Clusters of differentiation; TLR4: Toll-like receptor 4; NF-κB: Nuclear factor kappa B; IL: Interleukin; iNOS: Inducible nitric oxide synthase; ERK: Extracellular regulated protein kinases; MAPK: Mitogen activated protein kinase; AP-1: Amphipathic protein-1; COX2: Cyclooxygenase2; TNF-α: Tumor necrosis factor-α.

**Figure 3 foods-12-00320-f003:**
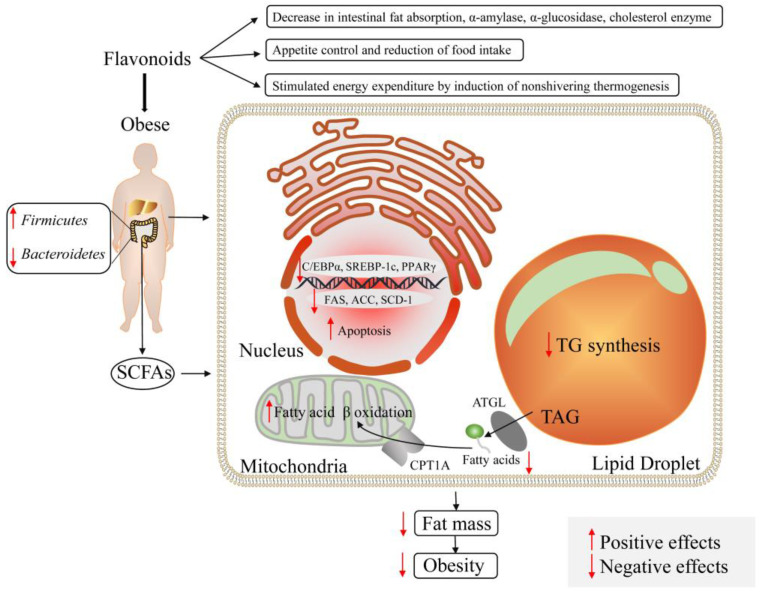
The main mechanisms of anti-obesity of flavonoids through modulating gut microbiota. C/EBP: CCAAT/enhancer-binding proteins, SREBP-1C: Sterol regulatory element binding protein-1c, PPARγ: Peroxisome proliferator-activated receptor γ, FAS: Fatty acid synthase, ACC: Acetyl-CoA carboxylase, SCD-1: Stearyl coenzyme A desaturated enzyme-1, ATGL: Adipose triglyceride lipase, CPT1A: Carnitine palmitoyltransferase 1A, TG: Triglyceride, TAG: Triacylglycerol, SCFAs: Short-chain fatty acids.

**Table 1 foods-12-00320-t001:** Classification of flavonoids.

Types	Bone	Sources of Plants	Main Flavonoids	Functions	References
Flavonols	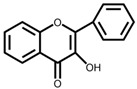	Dicotyledonous plants, onion, apples, broccoli, tea and red wine	Kaempferol, quercetin, prunetin, rutin	Anti-allergic, anti-cancer, antioxidant anti-inflammatory, the treatment of neurodegenerative diseases and gastrointestinal disorders. etc.	[8,9,10,11]
Isoflavones	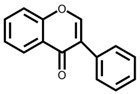	Legumes, Iridaceae, and Rosaceae, cereals, especially rye, oats, wine and tea	Daidzein, genisten, glycitein, biochain A, formononetein	Anti-cancer, antioxidant, anticareinogenic prevention of menopause, improvement of osteoporosis, improvement of diabetes, prevention of cardiovascular disease as well as menopausal symptoms, bone health	[12,13,14,15,16,17]
Flavanones	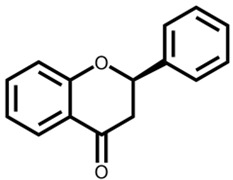	Citrus fruit and juice, orange peel, citron, blueberry extract, sweet root, red and blue flowers	Hesperetin, naringenin, eriodictyol, isosakuranetin and their glycosides	Anti-coagulation and hemostasis, anti-tumor activity, anti-cancer, antimutagenic, anti-oxidation, anti-bacterial, anti-inflammatory, prevention of cardiovascular disease and atherosclerosis	[18,19,20,21,22,23]
Anthocyanins	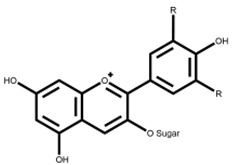	The cell sap of plant flowers, fruits, stems, leaves and roots	Pelargonidin, cyanidin, peonidin, delphinidin, petunidin and malvidin	Antioxidant, anti-tumor, anti-cancer, anti-inflammatory, inhibition of lipid peroxidation and platelet aggregation, prevention of diabetes, protection of eyesight	[24,25,26,27,28,29]
Flavones	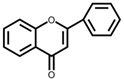	Angiosperms, mint family (*Lamiaceae*), cereals, and legumes (millet and sorghum)	Apigenin and luteolin flavone and their glycosides	Antioxidant, lowering cholesterol, inhibition of the activity of drug-metabolizing enzymes	[30,31,32]

**Table 2 foods-12-00320-t002:** Changes of intestinal flora in diseases.

Diseases Types		Alterations of Intestinal Flora	Reasons	References
Intestinal diseases	Acute and chronic diarrhea	Increase in external pathogens; Decrease in resident bacteria such as *Bacteroides*, *Bifidobacterium, Enterobacter*	Irritable stomach syndrome, acute bacterial dysentery post diarrhea, malabsorption syndrome	[60,61]
	Colitis	Increase in *Proteobacteria* and *Bacteroidetes;* Decrease in *Firmicutes*	Environmental factors, genetic factors, immune factors, microbial infection, etc. Dysfunction of spleen and stomach	[62,63]
	Irritable bowel syndrome (IBS)	Increase in *Proteus* and *Bacillus;* Decrease in *Firmicutes*, *Bacteroides* and *Trichospirillum*	Changes in diet, stress, and immune function, mucosal secretion, and increase in intestinal mucosal permeability lead to descending microbial diversity and stability	[64,65]
	Small intestinal bacterial Overgrowth (SIBO)	Increase in anaerobic bacteria	Intestinal stasis and obstruction of antibacterial mechanism	[66,67]
	Hepatitis, cirrhosis, liver cancer	Increase in *Enterobacteria, Enterococcus*, yeast; Decrease in *Bifidobacterium/Staphylococcus, Bifidobacterium*	Aerobic gram-negative bacilli proliferate, Gastrointestinal congestion, edema, age, HBV DNA levels, CHB clinical phases, family history of liver cancer and no antiviral treatment	[68,69]
	Acute necrotizing pancreatitis	Increase in *Escherichia coli;* Decrease in *Bifidobacterium* and *Lactobacilli*	Obstructive factors, blood circulation disorders, pancreatic trauma, bacterial and viral infections, metabolic diseases	[70,71]
	Intestinal bacterial translocation	Increased in *Aerobic bacteria* such as *Enterobacter* and *Enterococcus;* Decrease in intestinal obligate anaerobes such as *Bifidobacterium* and *Lactobacillus*	Overgrowth of intestinal bacteria and dysregulation of intestinal flora, reduced host immune function and disruption of the intestinal mucosal barrier	[72,73]
Metabolic diseases	Type 1 diabetes	Increase in *Bacteroides, Pseudobacterium ovatus; * Decreased in *Firmicutes*	T cells mediate islet β cell destruction, genetic susceptibilities	[74,75]
	Type 2 diabetes	Increase in *Firmicutes;* Decrease in *Bacteroides*	High-fat diet promotes auxin and insulin secretion, and intestinal microorganisms regulate and enhance the enteric-brain-islet β -cell axis through acetic acid secretion, β -cell dysfunction and insulin resistance	[76,77]
Neurological diseases	Autism spectrum disorder (ASD)	Increase in *Bacteroids, paracetobacters, Clostridium, Clostridium faecalis* and *Keratobacteria; * Decrease in Anti-inflammatory genus *Bifidobacterium*	Genetic and environmental factors	[78,79,80]
	Multiple sclerosis (MS)	Increase in the relative abundance of *Spore and Ruminococcus; * Decrease in the relative abundance of *Prevotella, p-hydroxybenzobacter, Adlerkreuzia, Collinella, Lactobacillus, symbacteria* and *Haemophilus*	Genetics, infection, environmental factors (e.g., smoking and Vitamin D deficiency)	[81,82,83]
	Parkinson’s disease (PD)	Increase in *Lactobacillaceae* family, *Akkermansia* sp., *Bifidiobacterium* sp., *Campylobacter, Delft, Haemophilus, Lunaropterus, Neisseria,* *Actinobacteria, V errucobacteria,* and *Prevotella, Brucella*; Decrease in *Blautia* sp. and the family *Lachnospiraceae*	Age, male gender and some environmental factors	[84,85,86,87]

**Table 3 foods-12-00320-t003:** The conversion reactions of flavonoids by gut microbiota.

Flavonoids Classes	Conversion Reaction	Enzymes	Species/Strain	Substrate	Products	References
Flavonols	Hydrolysis	β-glucosidase /sulfatase	*Escherichia coli*	Baicalin	Baicalein	[108,109]
	Deglycosylation	α-Rhamnosidase	*Lachnospiraceae* (*Lachnoclostridium,* and *Eisenbergiella*), *Enterobacteriaceae, Tannerellaceae 63* and *Erysipelotrichaceae* species	Rutin	Quercetin and quercetin-3-glucoside	[110,111]
Isoflavones	Deglycoslation	β-glucosidase	*Bifidobacterium Adolescentis*, *Bifidobacterium animalis subsp lactis*, *Bifidobacterium bifidum*	Daidzin, daidzein, glycitin and genistin	S-equol, delphinidin and cyanidin	[112,113]
Flavanones	Deglycosylation	β-glucosidase	*Bacteroides* ovatus ATCC 8483T, *B.* ovatus strain	Rutin	Quercetin	[114]
	C-ring cleavage	Flavanone reductase	(*Clostridium orbiscindens*) strains ATCC 49531, 257, 258 and 264	Quercetin	3,4-Dihydroxyphenylacetic acid	[115]
Anthocyanins	Deglycosylation	β-glucosidase	*Bifidobacterium* animalis ssp. Lactis (*Bifidobacterium* lactis) BB-12, *Lactobacillus* casei LC-01, *L.* plantarum IFPL722	Malvidin-3-O-glucoside	Gallic acid, homogentisic acid, syringic acid	[116]
Flavones	Demethylation	Rhamno glucosides, C-glycosyl	*Lactobacillus*, *Bifidobacterium*	Tangeretin	Tangeretin-O- glucuronides	[117]
	Dehydroxylation, deglycosylation, methylation, and acetylation	β-D-glucosidase	Human intestinal bacterium *Escherichia* sp. 4	Diosmetin-7-O-glucoside	Diosmetin, acacetin	[118]

**Table 4 foods-12-00320-t004:** Flavonoids regulate gut microbes and their functions.

Ranges of Flavonoids	Active Component	Pathways	Metabolites	Alteration of Gut Microbiota	Health Influence	References
isoflavone	Daidzein	Dihydrodaidzein and detrahydrodaidzein	Equol, O-demethylangolensin and the lignan enterolactone	Increase in *Lactobacillus mucosaeEPI2, enterococcus faecium, EPI, Veillonella. sp, train EP, HGH 6* and *Julong 732,* the *taxaPseudoflavonifractor, Dorea,* and *Lachnospiraceae incertae sedis*	Regulation of human obesity, sugar and lipid metabolism, weight loss and lipid lowering effect	[48,96,122,141,142,143,144]
Flavanonol	Dihydromyricetin	Reduction, and dihydroxylation	M1, M2, M3	Increase in *Bacteroidia, Betaproteobacteria* and *Alcaligenaceae;* Decrease in *Clostridia, Negativicutes, Veillonellaceae, Rikenellaceae, Peptococcaceae, Christensenellaceae* and *Ruminococcaceae*	Treatment of the diseases such as obesity, diabetes and atherosclerosis	[99,145]
Favonol	Quercetin	Heme oxygenase-1 (Hmox1, HO-1) dependent Pathway	3, 4-dihydroxyphenylacetic acid, 3-(3-hydroxyphenyl) propionic acid, protocatechuic acid	Increase in *Bacteroidetes/Firmicutes* and *Bacteroides; * Decrease in *Proteobacteria*, *Actinobacteria* and segmented filamentous bacteria	Prevention of Inflammatory bowel disease	[146]
Flavone	Luteolin	Activator protein (AP)-1 pathway, nuclear Factor kappa-light-chain-enhancer of activated B cells (NF-κB) pathway, and signal transducer and activator of transcription (STAT) 3 pathway	Luteolin glucuronides	Increase in *Lactobacilli*, *Bifidobacterial and Bacterodies;* Decrease in *Staphylococcus aureus, Salmonella typhimurium* and *Pathogenic Clostridium*	Alleviation of inflammation	[147,148,149]
Flavonone	Naringin	Reduction and hydrolysis	Naringenin and 3-(4-hydroxyphenyl) propionic acid	Increase in *Bifidobacterium catenulatum; * Decrease in *Enterococcus caccae*	Anti-cancer, anti-inflammation, and neuroprotection	[150,151]
	Hypericum perforatum L. extract	G protein-coupled receptors GPR43 and GPR4	Primary bile acids such as cholic acids and chenodeoxycholic acid	Increase in *Bacteroidetes, Elusimicrobia* and *Gemmatimonadetes;* Decrease in *Firmicutes*	Menopausal hypercholesterolemia as well as other menopausal symptoms, such as hot flashes and depression	[152]
	Flos Chrysanthemi (Luteolin-7-O-glucoside, luteolin, apigenin-7-O-glucoside, diosmetin-7-O-glucoside, quercetin and acacetin)	Hydrolysis, hydroxylation, acetylation, methylation, hydrogenation and deoxidation	A total of 32 metabolites	Increase in *Lactobacillus* and *Bifidobacterium;* Decrease in *Enterobacter, Enterococcus, Clostridium* and *Bacteroides*	Maintenance of human health and disease prevention such as anti-cancer	[153]

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
