# Peer review of "The Interaction between Flavonoids and Intestinal Microbes: A Review"

_foods, 2023, doi:10.3390/foods12020320_

Round 1
Reviewer 1 Report
The authors have written a narrative review that centers around microbiota-flavonoid interactions. The topic of the review is very simple but important from a pharmacological perspective. However, there are several similar reviews already available (many of which are more detailed) that undermine the importance of the current paper. The following corrections are needed to be made in order to improve the depth of discussion in the paper:
1. Since there are several similar papers available in the literature, it is not clear what is special about this paper. The following papers are of all similar to this paper:
https://pubmed.ncbi.nlm.nih.gov/34511152/
https://www.frontiersin.org/articles/10.3389/fnut.2021.798038/full
https://www.ncbi.nlm.nih.gov/pmc/articles/PMC8394324/
https://www.cambridge.org/core/journals/british-journal-of-nutrition/article/abs/interactions-between-dietary-flavonoids-and-the-gut-microbiome-a-comprehensive-review/1E2CB10A973E0BBF2FB6AE384A4B84E1
https://doi.org/10.1017/S0007114521003627
and many more…….
2. Throughout the manuscript, give emphasis only on the flavonoids, not on the polyphenols. Flavonoids are a part of polyphenols.
3. It doesn’t seem that the authors have done a good job in the literature review in related fields. References of several prior highly cited papers are missing in the present manuscript, like:
DOI: 10.1016/j.phrs.2019.104367
https://doi.org/10.1016/j.copbio.2019.12.018
4. Gut microbes can effectively metabolize flavonoids since these phytochemicals are considered xenobiotics. However, the microbial enzymatic machinery dedicated to phytochemical degradation remains underexplored in the paper. Read and cite concepts from these papers:
https://www.sciencedirect.com/science/article/pii/S2225411022000323
https://www.science.org/doi/10.1126/science.aag2770
4. In recent years, the concept that the interaction between gut microbes, metabolites, and gut metabolism impacts the overall health of the host, has emerged (https://www.science.org/doi/10.1126/science.aag2770). These concepts in relation to the phytochemical metabolites and gut microbiota interactions remain under-explored in the paper.
5. It is important to emphasize the effects of the phytochemicals on the mucosal immune system or on the mucosal metabolic processes that would also influence the phytochemical metabolism. For example, gut microbes and phytochemicals both affect the gut barrier (https://doi.org/10.1016/j.phrs.2020.105135), mucosal inflammation (DOI: 10.1080/10408398.2019.1570913), bile acid signaling (https://doi.org/10.3390/ijms21186495), etc. These aspects need to be discussed in detail in light of dietary flavonoids.
6. Dietary phytochemicals can influence disease outcomes by influencing the gut microbiota. Especially in case of chronic diseases like IBD and cancers (https://doi.org/10.3390/biology11050757), diabetes and obesity (DOI: 10.1155/2018/9734845), or other human diseases (DOI: 10.1016/j.biochi.2021.10.010) needs to be discussed in details by focusing on specific mechanisms.
7. Gut microbiota metabolize phytochemicals. But how dietary phytochemicals achieve health beneficial effects by favorably modulating gut microbes remains under-discussed.
8. The manuscript in many parts, contains unnecessary discussions on well-known and well-established facts. For example, Section 2 (general discussions on flavonoid), table 1, table 2, section 3 (overview of microbiota), and table 3, represents extremely general concepts. The authors need to shorten these sections by 2/3 rd. Focus more on the microbiota-phytochemical reciprocal interactions that affect health and disease.
9. In the future perspective part, the authors need to provide their own thoughts that would promote newer lines of research and provide newer insights to the readers.
10. A section needs to be added that describes the role of gut microbiota in dictating phytochemical bioavailability.
Reviewer 2 Report
The manuscript (foods-2077490) entitled “The interaction between flavonoids and intestinal microbes: a review” is by Hui-hui Xiong, et al.
1. Some language editing is needed. For example, in the first paragraph of the Introduction section, the last sentence stated as “The gut microbiota is modified by various factors, including intrinsic factors such as changes in its own functional structure…”. What does “its” mean here?
2. The title is too broad as there are a lot of flavonoids. Please focus on one or two types.
3. Table 1 is irrelevant to the topic as it is for the extraction methods.
4. Sections 2.2.2 to 2.2.6 such as Anti-oxidative effect are also irrelevant to the topic.
5. Section 3 can be shortened as all the roles of microbiota have been known and summarized by others. There is no need to repeat them. A couple of paragraphs is enough.
6. The title of Figure 1 is too general. Flavonoids cover a broad range of molecules with a variety of structures. Please narrow it down to the molecules that this figure applies to.
7. Please use a table to summarize reactions described in section 4.2 and specific molecules involved. Using flavonoids does not help the goal of this manuscript.
8. Hesperidin in line 583 should not be capitalized.
9. In Figure 2, please use Luteolin, not flavonoids.
10. In section 6.2, please focus on the interaction of flavonoids and microbiota, but not obesity and gut microbiota, which have been described in numerous papers as you have indicated in line 872. The same is true for the section 6.3.
Reviewer 3 Report
In this work authors review the interaction between flavonoids and intestinal microbiota. They describe in some detail the biological effects as well as the metabolic biotransformation pathways of flavonoids highlighting the mutual potential association of flavonoids and intestinal microorganisms. Indeed, in this comprehensive review authors present a variety of physiological activities of flavonoids which are metabolized, or biotransformed by gut microbiota, thus producing new metabolites that promote human health by a reciprocal modulation of the composition and structure of the intestinal microbiota. Authors are attempting to explain the bidirectional interaction between flavonoids and microbiota and thus provide theoretical basis for the promotion of gastrointestinal health and the prevention and treatment of some dysbiosis-associated diseases.
For publication purposes there are some issues the authors should address:
In general, the sentences are often very long or even incomprehensible. Moreover, different sentences are repeated. There are no transition sentences between the assertions or hypotheses and the results of the studies presented. These aspects should be reviewed.
In Table 2 (and related text) and in general it should be clarified that these are proposed or suggested "functions" for flavonoids.
Minor comments:
Line 74, write in full UHPLC-Q-TOF/MS, mentioned for the first time in the text..
Line 106, write in full RFTF, mentioned for the first time in the text.
Line 168, It should be written “It is internationally recognized that…”
Line 178, Redundancy. It should be written “…but still need more studies to prove these results”
Line 193, the title should say "the effects of flavonoids"
Line 254, Redundancy. Delete “… and cell cycle arrest.”
Paragraph 256-264, unclear text. Sentences should be better explained.
Line 266, clarify under what conditions daidzein and genistein are beneficial to preventing breast cancer risk.
Line 278, add species name to gender Staphylococcus CCMB 285
Line 301, clarify the opposite effect of Epimedium in promoting vascular smooth muscle cell apoptosis.
Lines 309-313, correct the incorrect sentence
Line 358, separate references 99 and 88
Line 360, complete the incomplete sentence.
Line 369, correct grammar.
Line 387, what the authors do.
Line 387, What do the authors mean by "the intestinal inflammation bacteria diversity"?
Line 398, add comma … mucin, peristalsis…
Line 412, complete “intestinal immune system”
Line 418-419, clarify the sentence
Line 428, Replace the word "severely"
Line 432, it should be written “However, more studies should be carried out to…”
Line 451, it should be written “Moreover, haw microbes metabolize flavonoids…”
Lines 464, 474, these are examples of a repeated sentences.
Line 479, it should be written “The metabolic 479 mechanisms of flavonoids by gut microbiota and impacts on health are shown in Figure 1.”
From line 468, it will be necessary to correct the sentences which begin with "And". Instead, it would be appropriate to introduce words like "de plus", "in addition", etc.
Line 506-507, what does "thick-walled phylum" and "anaphylactic phylum" mean?
Line 516, “Gut microbiota produces…”
Line 516-518, review the sentence.
Line 520, it should be written “are involved in the conversion of flavonoids.”
Lines 522-523, repeated sentence.
Line 534, method stated in full for the first time when it should appear on line 74.
Line 556, it should be added “Polymethoxyflavones (PMF)-metabolizing bacterium”
Lines 565-571, unclear sentence. Please reformulate.
Lines 574-578, sentence too long and difficult to understand.
Line 583, it should be added “Hesperedin bioavailability”
Line 585, an inconsistency is noticed. Clarify this sentence.
Line 593-594, reformulate this sentence.
Line 636, it should be added “The experiment reported by Wang et al.”
Line 638, correct “could not enter…”
Line 704, “A functional intestinal mucosal barrier”
Lines 720, 862, do not quote all the authors, but only the first and add "et al"
Line 725, What do the authors mean with the phrase "some limitations of flavonids"?
Line 742, phrase that repeats the title of the sub-chapter
Line 746, it should be written “Flavonoids are showed to achieve…”
Line 759, Write triple negative breast cancer (TNBC) in full
Lines 826-829, 846, 848, all bacterial genera should be written in italics and with the first letter capitalized
Line 891, in full PRDX4 (Peroxiredoxin 4)
Line 931, correct “fat physical activity”
Lines 935-938, redundant
Round 2
Reviewer 2 Report
The authors have addressed my comments.
Reviewer 3 Report
I believe the manuscript has been sufficiently improved to warrant publication in Foods.
